# OPTIMAL TRANSPORT FOR OFFLINE IMITATION LEARNING

**Yicheng Luo**
University College London

**Zhengyao Jiang**
University College London

**Samuel Cohen**
University College London

**Edward Grefenstette**
University College London & Cohere

**Marc Peter Deisenroth**
University College London

## ABSTRACT

With the advent of large datasets, offline reinforcement learning (RL) is a promising framework for learning good decision-making policies without the need to interact with the real environment. However, offline RL requires the dataset to be reward-annotated, which presents practical challenges when reward engineering is difficult or when obtaining reward annotations is labor-intensive. In this paper, we introduce Optimal Transport Reward labeling (OTR), an algorithm that assigns rewards to offline trajectories, with a few high-quality demonstrations. OTR's key idea is to use optimal transport to compute an optimal alignment between an unlabeled trajectory in the dataset and an expert demonstration to obtain a similarity measure that can be interpreted as a reward, which can then be used by an offline RL algorithm to learn the policy. OTR is easy to implement and computationally efficient. On D4RL benchmarks, we show that OTR with a single demonstration can consistently match the performance of offline RL with ground-truth rewards[1].

## 1 INTRODUCTION

Offline Reinforcement Learning (RL) has made significant progress recently, enabling learning policies from logged experience without any interaction with the environment. Offline RL is relevant when online data collection can be expensive or slow, e.g., robotics, financial trading and autonomous driving. A key feature of offline RL is that it can learn an improved policy that goes beyond the behavior policy that generated the data. However, offline RL requires the existence of a reward function for labeling the logged experience, making direct applications of offline RL methods impractical for applications where rewards are hard to specify with hand-crafted rules. Even if it is possible to label the trajectories with human preferences, such a procedure to generate reward signals can be expensive. Therefore, enabling offline RL to leverage unlabeled data is an open question of significant practical value.

Besides labeling every single trajectory, an alternative way to inform the agent about human preference is to provide expert demonstrations. For many applications, providing expert demonstrations is more natural for practitioners compared to specifying a reward function. In robotics, providing expert demonstrations is fairly common, and in the absence of natural reward functions, 'learning from demonstration' has been used for decades to find good policies for robotic systems; see, e.g., (Atkeson & Schaal, 1997; Abbeel & Ng, 2004; Calinon et al., 2007; Englert et al., 2013). One such framework for learning policies from demonstrations is imitation learning (IL). IL aims at learning policies that imitate the behavior of expert demonstrations without an explicit reward function.

There are two popular approaches to IL: Behavior Cloning (BC) (Pomerleau, 1988) and Inverse Reinforcement Learning (IRL) (Ng & Russell, 2000). BC aims to recover the demonstrator's behavior directly by setting up an offline supervised learning problem. If demonstrations are of high quality and actions of the demonstrations are recorded, BC can work very well as demonstrated by Pomerleau (1988), but generalization to new situations typically does not work well. IRL learns an intermediate reward function that aims to capture the demonstrator's intent. Current algorithms

---

[1]Code is available at `https://github.com/ethanluoyc/optimal_transport_reward`

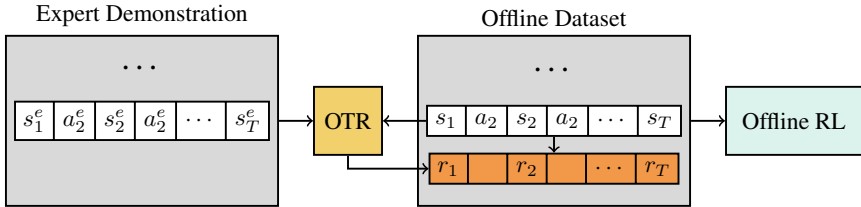

Figure 1: Illustration of Optimal Transport Reward Labeling (OTR). Given expert demonstrations (left) and an offline dataset without reward labels (center), OTR adds reward labels $r_i$ to the offline dataset by means of optimal transport (orange, center). The labeled dataset can then be used by an offline RL algorithm (right) to learn policies.

have demonstrated very strong empirical performance, requiring only a few expert demonstrations to obtain good performance (Ho & Ermon, 2016; Dadashi et al., 2022). While these IRL methods do not require many demonstrations, they typically focus on the *online* RL setting and require a large number of environment interactions to learn an imitating policy, i.e., these methods are not suitable for offline learning.

In this paper, we introduce Optimal Transport Reward labeling (OTR), an algorithm that uses optimal transport theory to automatically assign reward labels to unlabeled trajectories in an offline dataset, given one or more expert demonstrations. This reward-annotated dataset can then be used by offline RL algorithms to find good policies that imitate demonstrated behavior. Specifically, OTR uses optimal transport to find optimal alignments between unlabeled trajectories in the dataset and expert demonstrations. The similarity measure between a state in unlabeled trajectory and that of an expert trajectory is then treated as a reward label. These rewards can be used by any offline RL algorithm for learning policies from a small number of expert demonstrations and a large offline dataset. Figure 1 illustrates how OTR uses expert demonstrations to add reward labels to an offline dataset, which can then be used by an offline RL algorithm to find a good policy that imitates demonstrated behavior. Empirical evaluations on the D4RL (Fu et al., 2021) datasets demonstrate that OTR recovers the performance of offline RL methods with ground-truth rewards with only a single demonstration. Compared to previous reward learning and imitation learning approaches, our approach also achieves consistently better performance across a wide range of offline datasets and tasks.

## 2 OFFLINE REINFORCEMENT LEARNING AND IMITATION LEARNING

**Offline Reinforcement Learning**   In offline/batch reinforcement learning, we are interested in learning policies directly from fixed offline datasets (Lange et al., 2012; Levine et al., 2020), i.e., the agent is not permitted any additional interaction with the environment. Offline RL research typically assumes access to an offline dataset of observed transitions $\mathcal{D} = \{(s_t^i, a_t^i, r_t^i, s_{t+1}^i)\}_{i=1}^N$. This setting is particularly attractive for applications where there is previous logged experience available but online data collection is expensive (e.g., robotics, healthcare). Recently, the field of offline RL has made significant progress, and many offline RL algorithms have been proposed to learn improved policies from diverse and sub-optimal offline demonstrations (Levine et al., 2020; Fujimoto & Gu, 2021; Kumar et al., 2020; Kostrikov et al., 2022c; Wang et al., 2020).

Offline RL research typically assumes that the offline dataset is reward-annotated. That is, each transition $(s_t^i, a_t^i, r_t^i, s_{t+1}^i)$ in the dataset is labeled with reward $r_t^i$. One common approach to offline RL is to leverage the reward signals in the offline dataset and learn a policy with an actor-critic algorithm purely from offline transitions (Levine et al., 2020). However, in practice, having per-transition reward annotations for the offline dataset may be difficult due to the challenges in designing a good reward function. Zolna et al. (2020) propose ORIL which learns a reward function based on positive-unlabeled (PU) learning (Elkan & Noto, 2008) that can be used to add reward labels to offline datasets, allowing for unlabeled datasets to be used by offline RL algorithms.

**Imitation Learning** Imitation Learning (IL) aims at learning policies that can mimic the behavior of an expert demonstrator. Unlike the typical RL setting, a reward function is not required. Instead, IL methods assume access to an expert that can provide demonstrations of desired behavior.

There are two commonly used approaches to imitation learning. Behavior Cloning (BC) (Pomerleau, 1988) learns a policy by learning a mapping from states to actions in the trajectory demonstrated by an expert. By formulating policy learning as a supervised learning problem, BC can suffer from classical problems of supervised regression: generalizing the policy to unseen states may not be successful due to overfitting; multiple expert trajectories that follow different paths (bifurcation) will cause challenges. Inverse Reinforcement Learning (IRL) considers first learning a reward function based on expert demonstrations. The learned reward function can then be used to train a policy using an RL algorithm. Previous work shows that IRL-based methods can learn good policies with a small number of expert demonstrations. However, popular recent IRL methods, such as GAIL (Ho & Ermon, 2016), require a potentially large number of online samples during training, resulting in poor sample efficiency. Moreover, algorithms, such as GAIL, follow a training paradigm that is similar to Generative Adversarial Networks (GANs) (Goodfellow et al., 2014), by formulating the problem of IRL as a minimax optimization problem to learn a discriminator that implicitly minimizes an f-divergence. It was found, however, that Adversarial imitation learning (AIL) methods such as GAIL can be very difficult to optimize, requiring careful tuning of hyperparameters (Orsini et al., 2021). Another popular approach is to perform distribution matching between the policy and the expert demonstrations (Englert et al., 2013; Kostrikov et al., 2022b).

More recently, IRL methods based on Optimal Transport (OT) have demonstrated success as an alternative method for IRL compared to AIL approaches. Unlike AIL approaches, OT methods minimize the Wasserstein distance between the expert's and the agent's state-action distributions. Building on this formulation, Xiao et al. (2019) propose to minimize the Wasserstein distance via its dual formulation, which may lead to potential optimization issues. More recently, Dadashi et al. (2022) introduce PWIL, which instead minimizes the Wasserstein distance via its primal formulation, avoiding the potential optimization issues in the dual formulation. Along this line of work, Cohen et al. (2021) propose a series of improvements to the primal Wasserstein formulation used in PWIL and demonstrate strong empirical results in terms of both sample efficiency and asymptotic performance. Still, these approaches require a large number of online samples to learn good policies. While progress has been made to improve the sample efficiency of these approaches (Kostrikov et al., 2022a), imitation learning without any online interaction remains an active research area. In this paper, we provide a constructive algorithm to address the issue of offline imitation learning by using optimal transport to annotate offline datasets with suitable rewards.

## 3 Offline Imitation Learning with Optimal Transport

We consider learning in an episodic, finite-horizon Markov Decision Process (MDP) $(\mathcal{S}, \mathcal{A}, p, r, \gamma, p_0, T)$ where $\mathcal{S}$ is the state space, $\mathcal{A}$ is the action space, $p$ is the transition function, $r$ is the reward function, $\gamma$ is the discount factor, $p_0$ is the initial state distribution and $T$ is the episode horizon. A policy $\pi$ is a function from state to a distribution over actions. The goal of RL is to find policies that maximize episodic return. Running a policy $\pi$ in the MDP generates a state-action episode/trajectory $(s_1, a_1, s_2, a_2, \ldots, s_T) =: \tau$.

We consider the problem of imitation learning purely from offline datasets. Unlike in the standard RL setting, no explicit reward function is available. Let $\tau = (s_1, a_1, s_2, a_2, \ldots, s_T)$ denote an episode of interaction with the MDP using a policy that selects actions $a_t$ at time steps $t = 1, \ldots, T-1$. Instead of a reward function, we have access to a dataset of *expert* demonstrations $\mathcal{D}^e = \{\tau_e^{(n)}\}_{n=1}^N$ generated by an expert policy $\pi_e$ and a large dataset of *unlabeled* trajectories $\mathcal{D}^u = \{\tau_\beta^{(m)}\}_{m=1}^M$ generated by an arbitrary behavior policy $\pi_\beta$. We are interested in learning an offline policy $\pi$ combining information from the expert demonstrations and unlabeled experience, without any interaction with the environment. We will address this problem by using optimal transport, which will provide a way to efficiently annotate large offline RL datasets with rewards.

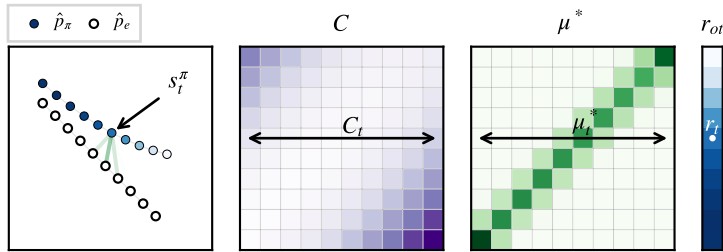

Figure 2: Illustration of the computations performed by OTR. In this example, we consider an MDP with a two-dimensional state-space ($|\mathcal{S}| = 2$). We have two empirical state distributions from an expert $\hat{p}_e$ with samples $\{s_{t'}^e\}_{t'=1}^{T'}$ (○) and policy $\hat{p}_\pi$ with samples $\{s_t^\pi\}_{t=1}^T$ (●) as denoted by points in the leftmost figure. OTR assigns rewards $r_{\text{ot}}$ (blue) to each sample in the policy's empirical state distribution as follows: (i) Compute the pairwise cost matrix $C$ (purple) between expert trajectories and trajectories generated by behavior policy; (ii) Solve for the optimal coupling matrix $\mu$ (green) between $\hat{p}_\pi$ and $\hat{p}_e$; (iii) Compute the reward for $s_t^\pi$ as $r_{\text{ot}}(s_t^\pi) = -C_t^T \mu_t^*$. Consider for example a state $s_t^\pi \in \hat{p}_\pi$; the row $C_t$ in the cost matrix corresponds to the costs between $s_t^\pi$ and $\{s_{t'}^e\}_{t'=1}^{T'}$. $\mu_t^*$ represents the optimal coupling between $s_t^\pi$ and the expert samples. The optimal coupling moves most of the probability mass to $s_3^e$ and a small fraction of the mass to $s_4^e$ (green lines in the leftmost figure).

### 3.1 Reward Labeling via Wasserstein Distance

Optimal Transport (OT) (Cuturi, 2013; Peyré & Cuturi, 2020) is a principled approach for comparing probability measures. The (squared) Wasserstein distance between two discrete measures $\mu_x = \frac{1}{T} \sum_{t=1}^T \delta_{x_t}$ and $\mu_y = \frac{1}{T'} \sum_{t=1}^{T'} \delta_{y_t}$ is

$$\mathcal{W}^2(\mu_x, \mu_y) = \min_{\mu \in M} \sum_{t=1}^T \sum_{t'=1}^{T'} c(x_t, y_{t'}) \mu_{t,t'}, \tag{1}$$

where $M = \{\mu \in \mathbb{R}^{T \times T'} : \mu\mathbf{1} = \frac{1}{T}\mathbf{1}, \mu^T\mathbf{1} = \frac{1}{T'}\mathbf{1}\}$ is the set of coupling matrices, $c$ is a cost function, and $\delta_x$ refers to the Dirac measure for $x$. The optimal coupling $\mu^*$ provides an alignment between the samples in $\mu_x$ and $\mu_y$. Unlike other divergence measures (e.g., KL-divergence), the Wasserstein distance is a metric and it incorporates the geometry of the space.

Let $\hat{p}_e = \frac{1}{T'} \sum_{t=1}^{T'} \delta_{s_t^e}$ and $\hat{p}_\pi = \frac{1}{T} \sum_{t=1}^T \delta_{s_t^\pi}$ denote the empirical state distribution of an expert policy $\pi_e$ and behavior policy $\pi$ respectively. Then the (squared) Wasserstein distance

$$\mathcal{W}^2(\hat{p}_\pi, \hat{p}_e) = \min_{\mu \in M} \sum_{t=1}^T \sum_{t'=1}^{T'} c(s_t^\pi, s_{t'}^e) \mu_{t,t'} \tag{2}$$

can be used to measure the distance between expert policy and behavior policy. Let $\mu^*$ denote the optimal coupling for the optimization problem above, then eq. (2) provides a reward signal

$$r_{\text{ot}}(s_t^\pi) = -\sum_{t'=1}^{T'} c(s_t^\pi, s_{t'}^e) \mu_{t,t'}^*, \tag{3}$$

which can be used for learning policy $\pi$ in an imitation learning setting.

### 3.2 Imitation Learning Using Reward Labels From Optimal Transport

We leverage the reward function from eq. (3) to annotate the unlabeled dataset with reward signals. Computing the optimal alignment between the expert demonstration with trajectories in the unlabeled dataset allows us to assign a reward for each step in the unlabeled trajectory. Figure 2 illustrates the computation performed by OTR to annotate an unlabeled dataset with rewards using demonstrations from an expert.

The pseudo-code for our approach is given in algorithm 1. OTR takes the unlabeled dataset $\mathcal{D}^u$ and expert demonstration $\mathcal{D}^e$ as input. For each unlabeled trajectory $\tau^{(m)} \in \mathcal{D}^u$, OTR solves the optimal

---

**Algorithm 1:** Pseudo-code for Optimal Transport Reward labeling (OTR)

    **Input:** unlabeled dataset $\mathcal{D}^u$, expert dataset $\mathcal{D}^e$
    **Output:** labeled dataset $\mathcal{D}^{\text{label}}$

**1** $\mathcal{D}^{\text{label}} \leftarrow \emptyset$
**2** **foreach** $\tau^{(m)}$ *in* $\mathcal{D}^u$ **do**               // Label each episode in the unlabeled dataset
**3**     $C^{(m)}, \mu^{*(m)} \leftarrow \texttt{SolveOT}(\mathcal{D}^e, \tau^{(m)})$    // Compute the optimal alignment with eq. (2)
**4**     **for** $t = 1$ *to* $T$ **do**
**5**         $r_{\text{OT}}(s_t^{(m)}) \leftarrow -\sum_{t'=1}^{T'} C_{t,t'}^{(m)} \mu_{t,t'}^{*(m)}$    // Compute the per-step rewards with eq. (3)
**6**     **end**
**7**     $\mathcal{D}^{\text{label}} \leftarrow \mathcal{D}^{\text{label}} \cup (s_1^{(m)}, a_1^{(m)}, r_1^{\text{OT}}, \ldots, s_T^{(m)})$    // Append labeled episode
**8** **end**
**9** **return** $\mathcal{D}^{label}$

---

transport problem for each, obtaining the cost matrix $C^{(m)}$ and optimal alignment $\mu^{*(m)}$ (line 3). OTR then computes the per-step reward label following eq. (3) (line 5). The reward-annotated trajectories are then combined, forming a reward-labeled dataset $\mathcal{D}^{\text{label}}$.

Solving eq. (2) requires solving the OT problem to obtain the optimal coupling matrix $\mu^*$. This amounts to solving a linear program (LP), which may be prohibitively expensive with a standard LP solver. In practice, we solve the entropy-regularized OT problem with Sinkhorn's algorithm (Cuturi, 2013). We leverage the Sinkhorn solver in OTT-JAX (Cuturi et al., 2022) for this computation. Once OTR has annotated the unlabeled offline dataset with intrinsic rewards, we can use an offline RL algorithm for learning a policy. Since we are working in the pure offline setting, it is important to use an offline RL algorithm that can minimize the distribution shifts typically encountered in the offline setting.

Unlike prior works that compute rewards using online samples (Dadashi et al., 2022; Cohen et al., 2021), we compute the rewards entirely offline, prior to running offline RL training, avoiding the need to modify any part of the downstream offline RL pipeline. Therefore, our approach can be combined with any offline RL algorithms, providing dense reward annotations that are required by the downstream algorithms. Figure 1 illustrates the entire pipeline of using OTR for relabeling and running an offline RL algorithm using the reward annotated datasets.

Compared to previous work that aims at solving offline imitation learning with a single algorithm, OTR focuses on generating high-quality reward annotations for downstream offline RL algorithms. In addition, our approach enjoys several advantages:

- Our approach does not require training separate reward models or discriminators, which may incur higher runtime overhead. By not having to train a separate parametric model, we avoid hyper-parameter tuning on the discriminator network architectures.

- Unlike other approaches, such as GAIL or DemoDICE, our approach does not require solving a minimax optimization problem, which can suffer from training instability (Orsini et al., 2021).

- Our approach is agnostic to the offline RL methods for learning the policy since OTR computes reward signals independently of the offline RL algorithm.

## 4 EXPERIMENTS

In this section, we evaluate OTR on D4RL Locomotion, Antmaze, and Adroit benchmark tasks (see also Figure 3) with the goal of answering the following research questions:

1. Can OTR recover the performance of offline RL algorithms that has access to a well-engineered reward function (i.e., ground-truth rewards provided by the environment)?

2. Can OTR handle unlabeled datasets with behaviors of unknown and mixed quality?

3. How does OTR perform with a varying number of expert demonstrations?

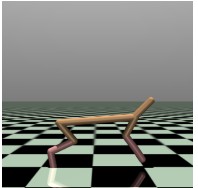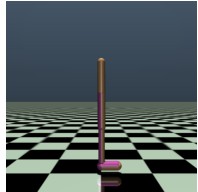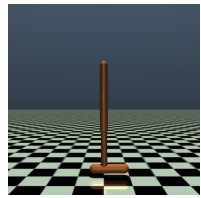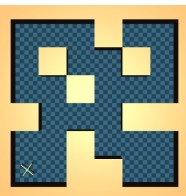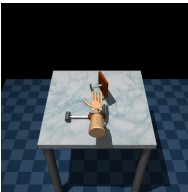

Figure 3: Benchmark tasks: D4RL Locomotion, Antmaze, and Adroit.

4. How does OTR compare with previous work on offline imitation learning in terms of performance and runtime complexity?

We demonstrate that OTR can be effectively combined with an offline RL algorithm to learn policies from a large dataset of unlabeled episodes and a small number of high-quality demonstrations. Since OTR is only a method for reward learning, it can be combined with any offline RL algorithm that requires reward-annotated data for offline learning. In this paper, we combine OTR with the Implicit $Q$-Learning (IQL) algorithm (Kostrikov et al., 2022c).

## 4.1 SETUP

We evaluate the performance of OTR+IQL on the D4RL locomotion benchmark (Fu et al., 2021). We start by evaluating OTR on three environments (`HalfCheetah-v2`, `Walker-v2`, `Hopper-v2`) from the OpenAI Gym MuJoCo locomotion tasks. For each environment, we use the `medium-v2`, `medium-replay-v2` and `medium-expert-v2` datasets to construct the expert demonstrations and the unlabeled dataset. For the expert demonstrations, we choose the best episodes from the D4RL dataset based on the episodic return.[2] To obtain the unlabeled dataset, we discard the original reward information in the dataset. We then run OTR to label the dataset based on the optimal coupling between the unlabeled episodes and the chosen expert demonstrations. Afterward, we proceed with running the offline RL algorithm.

For OTR, we follow the recommendation by Cohen et al. (2021) and use the cosine distance as the cost function. When there are more than one episode of expert demonstrations, we compute the optimal transport with respect to each episode independently and use the rewards from the expert trajectory that gives the best episodic return. Similar to (Dadashi et al., 2022; Cohen et al., 2021), we squash the rewards computed by line 5 with an exponential function $s(r) = \alpha \exp(\beta r)$. This has the advantage of ensuring that the rewards consumed by the offline RL algorithm have an appropriate range since many offline RL algorithms can be sensitive to the scale of the reward values. We refer to appendix A.1 for additional experimental details and hyperparameters.

**Implementation** We implement OTR in JAX (Bradbury et al., 2018). For computing the optimal coupling, we use OTT-JAX (Cuturi et al., 2022), a library for optimal transport that includes a scalable and efficient implementation of the Sinkhorn algorithm that can leverage accelerators, such as GPU or TPU, for speeding up computations. Our IQL implementation is adapted from (Kostrikov et al., 2022c)[3], and we set all hyperparameters to the ones recommended in the original paper. All of our algorithms and baselines are implemented in Acme (Hoffman et al., 2022).

---

[2]In practice, the expert demonstrations can be provided separately; we only select the expert demonstration in this way for ease of evaluation.

[3]https://github.com/ikostrikov/implicit_q_learning

Our implementation is computationally efficient[4], requiring only about 1 minute to label a dataset with 1 million transitions (or 1000 episodes of length 1000)[5]. For larger-scale problems, OTR can be scaled up further by processing the episodes in the dataset in parallel. Our implementation of OTR and re-implementation of baselines are computationally efficient. Even so, the training time for IQL is about 20 minutes, so that OTR adds a relatively small amount of overhead for reward annotation to an existing offline RL algorithm.

**Baselines** We compare OTR+IQL with the following baselines:

- IQL (oracle): this is the original Implicit $Q$-learning (Kostrikov et al., 2022c) algorithm using the ground-truth rewards provided by the D4RL datasets.

- DemoDICE: an offline imitation learning algorithm proposed by Kim et al. (2022). DemoDICE was found to perform better than previous imitation learning algorithms (e.g., ValueDICE (Kostrikov et al., 2022b)) that can also (in principle) work in the pure offline setting. We ran the original implementation[6] under the same experimental setting as we used for the other algorithms in the paper.

- ORIL: a reward function learning algorithm from (Zolna et al., 2020). ORIL learns a reward function by contrasting the expert demonstrations with the unlabeled episodes. The learned reward function is then used for labeling the unlabeled dataset. We implement ORIL in a comparable setting to the other baselines.

- UDS: we keep the rewards from the expert demonstrations and relabel all rewards from the unlabeled datasets with the minimum rewards from the environment. This was found to perform well in (Yu et al., 2022). This also resembles online imitation learning methods such as SQIL (Reddy et al., 2022) or offline RL algorithms, such as COG (Singh et al., 2021).

Since UDS and ORIL are also agnostic about the underlying offline RL algorithm used, we combined these algorithms with IQL so that we can focus on comparing the performance difference due to different approaches used in generating reward labels.

For all algorithms, we repeat experiments with 10 random seeds and report the mean and standard deviation of the normalized performance of the last 10 episodes of evaluation. We compare all algorithms by using either $K = 1$ or $K = 10$ expert demonstrations. Obtaining the results on the locomotion datasets took approximately 500 GPU hours.

## 4.2 RESULTS

**MuJoCo Locomotion** Table 1 compares the performance between OTR+IQL with the other baselines. Overall, OTR+IQL performs best compared with the other baselines in terms of aggregate score over all of the datasets we used, recovering the performance of IQL with ground-truth rewards provided by the dataset. While we found that other baselines can perform well on some datasets, the performance is not consistent across the entire dataset and can deteriorate significantly on some datasets. In contrast, OTR+IQL is the only method that consistently performs well for all datasets of different compositions.

---

[4]An efficient implementation of OTR is facilitated by JAX, which includes useful functionality that allows us to easily parallelize computations. Concretely, we JIT-compile the computation of rewards for one episode and further leverage the `vmap` function to compute the optimal coupling between an unlabeled episode with all of the expert episodes in parallel. Efficiently parallelizing the computation of the optimal coupling requires that all the episodes share the same length. This is necessary both for parallelizing the computation across multiple expert demonstrations as well as for avoiding recompilation by XLA due to changes in the shape of the input arrays. To achieve high throughput for datasets with varying episodic length, we pad all observations to the maximum episode length allowed by the environment (which is 1000 for the OpenAI Gym environments) but set the weights of the observations to zero. Padding the episodes this way does not change the solution to the optimal coupling problem. Note that padding means that a 1M transition dataset may create more than 1000 episodes of experience, in this case, the runtime for our OTR implementation may be higher effectively due to having to process a larger number of padded episodes.

[5]Runtime measured on `halfcheetah-medium-v2` with an NVIDIA GeForce RTX 3080 GPU.

[6]`https://github.com/geon-hyeong/imitation-dice`

| Dataset | BC | 10%BC | IQL (oracle) | DemoDICE | IQL+ORIL | IQL+UDS | OTR+IQL (ours) |
|---|---|---|---|---|---|---|---|
| halfcheetah-medium-v2 | 42.6 | 42.5 | 47.4 ± 0.2 | 42.5 ± 1.7 | **49.0 ± 0.2** | 42.4 ± 0.3 | 43.3 ± 0.2 |
| hopper-medium-v2 | 52.9 | 56.9 | 66.2 ± 5.7 | 55.1 ± 3.3 | 47.0 ± 4.0 | 54.5 ± 3.0 | **78.7 ± 5.5** |
| walker2d-medium-v2 | 75.3 | 75.0 | 78.3 ± 8.7 | 73.4 ± 2.6 | 61.9 ± 6.6 | 68.9 ± 6.2 | **79.4 ± 1.4** |
| halfcheetah-medium-replay-v2 | 36.6 | 40.6 | 44.2 ± 1.2 | 38.1 ± 2.7 | **44.1 ± 0.6** | 37.9 ± 2.4 | 41.3 ± 0.6 |
| hopper-medium-replay-v2 | 18.1 | 75.9 | 94.7 ± 8.6 | 39.0 ± 15.4 | 82.4 ± 1.7 | 49.3 ± 22.7 | **84.8 ± 2.6** |
| walker2d-medium-replay-v2 | 26.0 | 62.5 | 73.8 ± 7.1 | 52.2 ± 13.1 | **76.3 ± 4.9** | 17.7 ± 9.6 | 66.0 ± 6.7 |
| halfcheetah-medium-expert-v2 | 55.2 | 92.9 | 86.7 ± 5.3 | **85.8 ± 5.7** | **87.5 ± 3.9** | 63.0 ± 5.7 | **89.6 ± 3.0** |
| hopper-medium-expert-v2 | 52.5 | 110.9 | 91.5 ± 14.3 | **92.3 ± 14.2** | 29.7 ± 22.2 | 53.9 ± 2.5 | **93.2 ± 20.6** |
| walker2d-medium-expert-v2 | 107.5 | 109.0 | 109.6 ± 1.0 | **106.9 ± 1.9** | **110.6 ± 0.6** | 107.5 ± 1.7 | **109.3 ± 0.8** |
| locomotion-v2-total | 466.7 | 666.2 | 692.4 | 585.3 | 588.5 | 494.9 | 685.5 |
| runtime | 10m | 10m | 20m | 100m[*] | 30m | 20m | 22m |

[*] The runtime is measured with the original PyTorch implementation.

Table 1: D4RL performance comparison between IQL with ground-truth rewards and OTR+IQL with a single expert demonstration ($K = 1$). We report mean ± standard deviation per task and aggregate performance and highlight near-optimal performance in **bold** and extreme negative outliers in red. OTR+IQL is the only algorithm that performs consistently well across all domains.

**Runtime**  Despite applying optimal transport, we found that with a GPU-accelerated Sinkhorn solver (Cuturi et al., 2022), combined with our efficient implementation in JAX, OTR achieves a faster runtime compared to algorithms that learn additional neural networks as discriminators (DemoDICE (Kim et al., 2022)) or reward models (ORIL (Zolna et al., 2020)). For methods that learn a neural network reward function, an overhead of at least 10 minutes is incurred, whereas OTR only incurs approximately 2 minutes of overheads when compared with the same amount of expert demonstrations.

**Effect of the number of demonstrations**  We investigate if the performance of the baselines can be improved by increasing the number of expert demonstrations used. Table 2 compares the aggregate performance on the locomotion datasets between OTR and the baselines when we increase the number of demonstrations from $K = 1$ to $K = 10$. DemoDICE's performance improves little with the additional amount of expert demonstrates. While ORIL and UDS enjoy a relatively larger improvement, they are still unable to match the performance of IQL (oracle) or OTR in terms of aggregate performance despite using the same offline RL backbone. OTR's performance is close to IQL (oracle) even when $K = 1$ and matches the performance of IQL (oracle) with $K = 10$.

| locomotion-v2-total | $K = 1$ | $K = 10$ |
|---|---|---|
| DemoDICE | 585.3 | 589.3 |
| IQL+ORIL | 588.5 | 618.3 |
| IQL+UDS | 494.9 | 575.8 |
| OTR+IQL | **685.5** | **694.3** |
| IQL (oracle) | | 692.4 |

Table 2: Aggregate performances of different reward labeling algorithms with different numbers of expert demonstrations. OTR is the only algorithm that leads to an offline RL performance close to using ground-truth rewards.

**Antmaze & Adroit**  We additionally evaluate OTR+IQL on the `antmaze-v0` and `adroit-v0` datasets. Table 3 shows that OTR+IQL again recovers the performance of IQL with ground-truth rewards. This suggests that OTR+IQL can learn from datasets with diverse behavior and human demonstrations even without ground-truth reward annotation; additional results in appendix A.2.

**Qualitative comparison of the reward predictions**  Figure 4a provides a qualitative comparison of the reward predicted by OTR, ORIL, and UDS. UDS annotates all transitions in the unlabeled dataset with the minimum reward from the environment. Thus, the episodes with non-zero rewards are expert demonstrations. This means that UDS is unable to distinguish between episodes in the unlabeled datasets. Compared to reward learning algorithms, such as ORIL, OTR's reward prediction more strongly correlates with the ground-truth rewards from the environment, which is a good precondition for successful policy learning by downstream offline RL algorithm. We also evaluate OTR's reward prediction on more diverse datasets, such as those in antmaze. Figure 4b shows the expert demonstrations we used in `antmaze-medium-play-v0` (left) and the trajectories that received the best OTR reward labels in the unlabeled dataset (right). OTR correctly assigns more rewards to trajectories that are closer to the expert demonstrations.

| Dataset | IQL (oracle) | OTR+IQL |
|---|---|---|
| antmaze-large-diverse-v0 | 47.5 ± 9.5 | 45.5 ± 6.2 |
| antmaze-large-play-v0 | 39.6 ± 5.8 | 45.3 ± 6.9 |
| antmaze-medium-diverse-v0 | 70.0 ± 10.9 | 70.4 ± 4.8 |
| antmaze-medium-play-v0 | 71.2 ± 7.3 | 70.5 ± 6.6 |
| antmaze-umaze-diverse-v0 | 62.2 ± 13.8 | 68.9 ± 13.6 |
| antmaze-umaze-v0 | 87.5 ± 2.6 | 83.4 ± 3.3 |
| antmaze-v0-total | 378.0 | 384.0 |

| Dataset | IQL (oracle) | OTR+IQL |
|---|---|---|
| door-cloned-v0 | 1.60 | 0.01 ± 0.01 |
| door-human-v0 | 4.30 | 5.92 ± 2.72 |
| hammer-cloned-v0 | 2.10 | 0.88 ± 0.30 |
| hammer-human-v0 | 1.40 | 1.79 ± 1.43 |
| pen-cloned-v0 | 37.30 | 46.87 ± 20.85 |
| pen-human-v0 | 71.50 | 66.82 ± 21.18 |
| relocate-cloned-v0 | -0.20 | -0.24 ± 0.03 |
| relocate-human-v0 | 0.10 | 0.11 ± 0.10 |
| adroit-v0-total | 118.1 | 122.16 |

Table 3: Performance of OTR+IQL on `antmaze` and `adroit` with a single expert demonstration. Similar to the results on locomotion, OTR+IQL recovers the performance of offline RL with ground-truth rewards. We report mean ± standard deviation per task and aggregate performance.

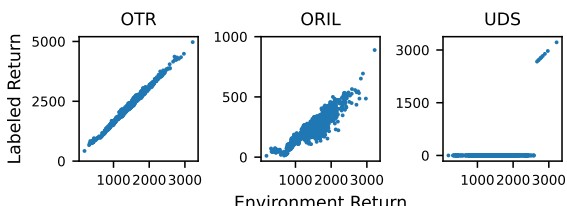 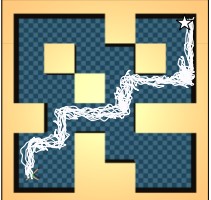 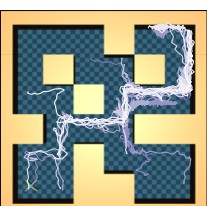

(a) Ground-truth return vs labeled return.

(b) Top trajectories selected by OTR.

Figure 4: Visualization of rewards predicted by OTR and baselines. (a) Qualitative differences between the rewards predicted by OTR, ORIL and UDS on `hopper-medium-v2`. (b) Visualization of top trajectories selected by OTR on `antmaze-medium-play-v0`. Left: Expert demonstrations. Right: ranking of trajectories according to rewards per step computed by OTR. Trajectories with lighter colors have higher rewards per step.

**Summary**    Our empirical evaluation demonstrates that OTR+IQL can recover the performance of offline RL with ground-truth rewards given only a single episode of expert demonstration. Compared to previous work, it achieves better performance even with a larger number of demonstrations. Our qualitative comparison shows that OTR assigns better reward estimates that correlate strongly with a hand-crafted reward function.

## 5   DISCUSSION

OTR can be effective in augmenting unlabeled datasets with rewards for use by downstream offline RL algorithms (see section 4). Compared to prior imitation learning and reward learning approaches, OTR enjoys better and more robust performance on the benchmarks we have considered in this paper. Furthermore, our approach is easy to implement. Most of the complexity of our approach arises from solving the optimal transport problem. However, there are libraries for solving OT efficiently in different frameworks or programming languages[7].

Unlike prior works, such as ValueDICE or DemoDICE, we split offline imitation learning into the two distinct phases of reward modeling via OTR and a subsequent offline RL. This improves modularity, allows for improvements in the two phases to be made independently, and adds flexibility. However, this modularity means that a practitioner needs to make a separate choice for the downstream offline RL algorithm.

We believe that OTR is most useful in situations where providing ground-truth rewards is difficult but providing good demonstrations and unlabeled data is feasible. However, it will not be feasible in cases where collecting expert demonstrations is difficult.

---

[7]For example, see Python Optimal Transport (POT) (Flamary et al., 2021), which supports PyTorch, JAX, or TensorFlow.

The Wasserstein distance formulation we used can be extended to perform cross-domain imitation learning by using the Gromov–Wasserstein distance to align expert demonstrations and offline trajectories from different spaces (Fickinger et al., 2022).

## 6 CONCLUSION

We introduced Optimal Transport Reward labeling (OTR), a method for adding reward labels to an offline dataset, given one or more expert demonstrations. OTR computes Wasserstein distances between expert demonstrations and trajectories in a dataset without reward labels, which then are turned into a reward signal. The reward-annotated offline dataset can then be used by an(y) offline RL algorithm to determine good policies. OTR adds minimal overhead to existing offline RL algorithms while providing the same performance as learning with a pre-specified reward function.

### ACKNOWLEDGMENTS

Yicheng Luo is supported by the UCL Overseas Research Scholarship (UCLORS). Samuel Cohen is supported by the UKRI AI Centre for Doctoral Training in Foundational Artificial Intelligence (EP/S021566/1).

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

# A APPENDIX

## A.1 HYPERPARAMETERS

Table 4 lists the hyperparameters used by OTR and IQL on the locomotion datasets. For Antmaze and Adroit, unless otherwise specified by table 5 or table 6, the hyperparameters follows from those used in the locomotion datasets.

The IQL hyperparameters are kept the same as those used in (Kostrikov et al., 2022c). Note that IQL rescales the rewards in the dataset so that the same set of hyperparameters can be used for datasets of different qualities. Since OTR computes rewards offline, we also apply reward scaling as in IQL. For the locomotion datasets, the rewards are rescaled by $\frac{1000}{\text{max\_return}-\text{min\_return}}$ while for antmaze we subtract 2 to the rewards computed by OTR. The reward processing in antmaze is different from the one used by the original IQL paper (which subtracts 1) since the rewards computed by OTR have a different range.

The squashing function used by OTR is based on the one used in (Dadashi et al., 2022). The antmaze squashing differs slightly from the one used in locomotion and adroit due to use of an earlier configuration. In practice, this should have minimal effect on the performance.

|  | Hyperparameter | Value |
|---|---|---|
|  | Discount | 0.99 |
| Network Architectures | Hidden layers | $(256, 256)$ |
|  | Dropout | none |
|  | Network initialization | orthogonal |
| IQL | Optimizer | Adam |
|  | Policy learning rate | $3e^{-4}$, cosine decay to 0 |
|  | Critic learning rate | $3e^{-4}$ |
|  | Value learning rate | $3e^{-4}$ |
|  | Target network update rate | $5e^{-3}$ |
|  | Temperature | 3.0 |
|  | Expectile | 0.7 |
| OTR | Episode length $T$ | 1000 |
|  | Cost function | cosine |
|  | Squashing function | $s(r) = 5.0 \cdot \exp(5.0 \cdot T \cdot r/|\mathcal{A}|)$ |

Table 4: OTR hyperparameters for D4RL Locomotion.

|  | Hyperparameter | Value |
|---|---|---|
| IQL | Temperature | 10.0 |
|  | Expectile | 0.9 |
| OTR | Squashing function | $s(r) = 5.0 \cdot \exp(T \cdot r)$ |

Table 5: OTR hyperparameters for D4RL Antmaze.

|  | Hyperparameter | Value |
|---|---|---|
| Network Architectures | Dropout | 0.1 |
| IQL | Temperature | 0.5 |
|  | Expectile | 0.7 |

Table 6: OTR hyperparameters for D4RL Adroit.

## A.2 ADDITIONAL EXPERIMENTAL RESULTS ON ADROIT AND ANTMAZE

We evaluate OTR on additional datasets from the antmaze and adroit domains with varying number of expert demonstrations. The results are presented in table 7 and table 8. OTR consistently recovers the performance of IQL with ground-truth rewards on these datasets, largely independent of the number $K$ of expert demonstrations provided.

| Dataset | IQL (oracle) | $K = 1$ OTR+IQL | $K = 10$ OTR+IQL |
|---|---|---|---|
| door-cloned-v0 | 1.60 | 0.01±0.01 | 0.01±0.01 |
| door-human-v0 | 4.30 | 5.92±2.72 | 4.15±2.08 |
| hammer-cloned-v0 | 2.10 | 0.88±0.30 | 1.31±0.70 |
| hammer-human-v0 | 1.40 | 1.79±1.43 | 1.36±0.22 |
| pen-cloned-v0 | 37.30 | 46.87±20.85 | 42.68±24.98 |
| pen-human-v0 | 71.50 | 66.82±21.18 | 69.41±21.50 |
| relocate-cloned-v0 | -0.20 | -0.24±0.03 | -0.24±0.03 |
| relocate-human-v0 | 0.10 | 0.11±0.10 | 0.10±0.07 |
| adroit-v0-total | 118.1 | 122.16 | 118.78 |

Table 7: OTR+IQL Results on Adroit. The standard deviations for IQL (oracle) are not available from (Kostrikov et al., 2022c).

| Dataset | IQL (oracle) | OTR+IQL ($K = 1$) | OTR+IQL ($K = 10$) |
|---|---|---|---|
| antmaze-large-diverse-v0 | 47.5±9.5 | 45.5±6.2 | 50.7±6.9 |
| antmaze-large-play-v0 | 39.6±5.8 | 45.3±6.9 | 51.2±7.1 |
| antmaze-medium-diverse-v0 | 70.0±10.9 | 70.4±4.8 | 70.5±6.9 |
| antmaze-medium-play-v0 | 71.2±7.3 | 70.5±6.6 | 72.7±6.2 |
| antmaze-umaze-diverse-v0 | 62.2±13.8 | 68.9±13.6 | 64.4±18.2 |
| antmaze-umaze-v0 | 87.5±2.6 | 83.4±3.3 | 88.7±3.5 |
| antmaze-v0-total | 378.0 | 384.0 | 398.2 |

Table 8: OTR+IQL Results on Antmaze.

## A.3 Combining OTR with Different Offline RL Algorithms

In the main experiments, we evaluated OTR by pairing it with the IQL algorithm. In this section, we investigate if OTR can recover the performance of a different offline RL algorithm (TD3-BC) (Fujimoto & Gu, 2021) using ground-truth rewards. We observe that (i) the performance from OTR+TD3-BC mostly matches those using the ground-truth rewards; (ii) the performance is fairly robust to the choice of the number of expert trajectories ($K = 1$ and $K = 10$ many expert demonstrations provide comparable performance). However, There are more variances on some datasets (e.g., `halfcheetah-medium-expert-v2`). Nevertheless, the differences are smaller compared to the baselines and OTR+TD3-BC still performs better than the baselines presented in section 4 in terms of the aggregate performance.

| Dataset | TD3-BC (oracle) | OTR+TD3-BC (K=1) | OTR+TD3-BC (K=10) |
|---|---|---|---|
| halfcheetah-medium-expert-v2 | 93.5±2.0 | 74.8±20.1 | 71.6±23.1 |
| halfcheetah-medium-replay-v2 | 44.4±0.8 | 39.4±1.3 | 38.9±1.5 |
| halfcheetah-medium-v2 | 48.0±0.7 | 42.6±1.0 | 42.7±1.1 |
| hopper-medium-expert-v2 | 100.2±20.0 | 103.2±13.9 | 98.9±19.7 |
| hopper-medium-replay-v2 | 64.8±25.5 | 74.9±28.8 | 80.2±23.1 |
| hopper-medium-v2 | 60.7±12.5 | 66.4±10.3 | 69.8±13.9 |
| walker2d-medium-expert-v2 | 109.5±0.5 | 109.0±0.6 | 108.8±0.8 |
| walker2d-medium-replay-v2 | 87.4±8.4 | 69.7±16.4 | 67.4±20.6 |
| walker2d-medium-v2 | 83.7±5.3 | 76.9±5.4 | 78.0±2.6 |
| locomotion-v2-total | 692.3 | 656.9 | 656.4 |

Table 9: OTR+TD3-BC Results on MuJoCo.

A.4    IMPORTANCE OF USING THE OPTIMAL TRANSPORT PLAN

In the main experiments, we compute the rewards based on the optimal coupling computed by the Sinkhorn solver. The optimal transport plan is sparse and transports most of the probability masses to only a few expert samples. In this section, we investigate what happens if we use a suboptimal transport plan where each sample from the policy's trajectory is transported equally to each sample in the expert's trajectory. In this case, the reward function essentially boils down to computing the average costs with respect to all of the states in the expert's trajectory.

| Dataset | OTR+IQL | OTR+IQL (Uniform Plan) |
|---|---|---|
| halfcheetah-medium-v2 | 43.3±0.2 | 43.5±0.3 |
| hopper-medium-v2 | 78.7±5.5 | 80.5±2.3 |
| walker2d-medium-v2 | 79.4±1.4 | 77.6±1.5 |
| halfcheetah-medium-replay-v2 | 41.3±0.6 | 41.6±0.8 |
| hopper-medium-replay-v2 | 84.8±2.6 | 69.8±10.1 |
| walker2d-medium-replay-v2 | 66.0±6.7 | 62.2±14.4 |
| halfcheetah-medium-expert-v2 | 89.6±3.0 | 90.6±2.9 |
| hopper-medium-expert-v2 | 93.2±20.6 | 89.2±14.0 |
| walker2d-medium-expert-v2 | 109.3±0.8 | 106.0±5.9 |

Table 10: OTR with Uniform Transport Plan

Table 10 compares the performance of OTR+IQL using the optimal transport plan and uniform transport plan. We find that for many datasets, using the suboptimal uniform transport plan is sufficient for reaching good performance. This indicates that using a reward function based on the similarity of states from the policy and the expert can be a simple and effective method for reward labeling. However, note that the uniform transport plan can still underperform compared to using the optimal transport plan (e.g., `hopper-medium-replay-v2`). This shows that the optimal transport formulation enables better and more consistent performance.

A.5    COMPARISON TO PWIL

In this section, we investigate if the online imitation learning algorithm PWIL (Dadashi et al., 2022) can be used in the offline setting with a change from using an online RL algorithm to an offline RL algorithm. We ran PWIL with IQL similar to what we did for OTR in the main paper. We use the PWIL implementation from Acme (Hoffman et al., 2022)[8].

Note that although OTR is similar to PWIL in using the Wasserstein distance to construct RL reward signals, OTR differs from PWIL in the choices of OT solver, the cost function as well as the approach used for aggregating results multiple expert demonstrations. Also note that for all experiments in the paper we consider learning only from expert state instead of state-action pairs. This is both a more general and challenging setting. It was found in (Dadashi et al., 2022) that PWIL sometimes perform badly without expert actions. We ran OTR and PWIL using only expert observations (denoted as OTR-state and PWIL-state) and OTR and PWIL using state-action pairs (denoted as OTR-action and PWIL-action). The results are illustrated in table 11. We found that we are unable to get good results when running PWIL using only expert state sequences. This is possibly due to difference choices of OT solver and cost functions. PWIL can perform well when combined with IQL to learn in the offline setting although sometimes performance is significantly worse compared to IQL oracle or OTR (e.g., `hopper-medium-expert-v2`).

In addition, Dadashi et al. (2022) found that PWIL's performance may deteriorate when learning from demonstrations consisting of only expert observations (i.e., no actions are present in the expert demonstrations).

---

[8]`https://github.com/deepmind/acme/tree/master/acme/agents/jax/pwil`

| K | 10 | | | |
| method | OTR-state | PWIL-state | OTR-action | PWIL-action |
|---|---|---|---|---|
| halfcheetah-medium-v2 | 43.1±0.3 | 1.6±1.2 | 43.4±0.3 | 47.5±0.2 |
| hopper-medium-v2 | 80.0±5.2 | 2.1±1.3 | 75.4±4.6 | 70.4±4.2 |
| walker2d-medium-v2 | 79.2±1.3 | 0.9±1.3 | 79.7±1.2 | 81.9±1.0 |
| halfcheetah-medium-replay-v2 | 41.6±0.3 | -2.3±0.5 | 41.9±0.3 | 44.6±1.1 |
| hopper-medium-replay-v2 | 84.4±1.8 | 1.4±1.2 | 85.3±1.1 | 89.7±4.9 |
| walker2d-medium-replay-v2 | 71.8±3.8 | -0.1±0.2 | 69.1±4.6 | 72.2±10.6 |
| halfcheetah-medium-expert-v2 | 87.9±3.4 | -0.3±1.5 | 88.3±5.1 | 88.6±4.3 |
| hopper-medium-expert-v2 | 96.6±21.5 | 1.5±0.6 | 86.6±22.9 | 32.9±25.0 |
| walker2d-medium-expert-v2 | 109.6±0.5 | 1.0±1.9 | 109.2±0.5 | 110.2±0.2 |

Table 11: Comparison between OTR and PWIL with IQL as offline RL backbone.

## A.6 HYPER-PARAMETER SENSITIVITY

| K | 10 | |
| method | OTR ($\alpha = \beta = 5$) | OTR ($\alpha = \beta = 1$) |
|---|---|---|
| halfcheetah-medium-expert-v2 | 87.9±3.4 | 86.9±4.0 |
| halfcheetah-medium-replay-v2 | 41.6±0.3 | 40.4±1.3 |
| halfcheetah-medium-v2 | 43.1±0.3 | 42.7±0.4 |
| hopper-medium-expert-v2 | 96.6±21.5 | 82.6±9.9 |
| hopper-medium-replay-v2 | 84.4±1.8 | 71.2±15.2 |
| hopper-medium-v2 | 80.0±5.2 | 75.7±6.4 |
| walker2d-medium-expert-v2 | 109.6±0.5 | 106.3±8.2 |
| walker2d-medium-replay-v2 | 71.8±3.8 | 63.2±5.7 |
| walker2d-medium-v2 | 79.2±1.3 | 77.4±1.5 |

Table 12: Effect of $\alpha$ and $\beta$ in the squashing function.

For the main results, the hyper-parameters for the squashing function ($\alpha$ and $\beta$) was chosen to be consistent with those used in (Dadashi et al., 2022). In this section we compare the differences in the choices of these hyper-parameters by running OTR with $\alpha = \beta = 1$. This reduces to simply applying an exponential transformation to the optimal transport costs. The results are illustrated in table 12. We find that OTR still performs well, demonstrating that it is not sensitive to the choices of these hyper-parameters.

