# OpenReview forum: "Optimal Transport for Offline Imitation Learning"
_ICLR.cc/2023/Conference — ICLR 2023 notable top 25%_

### Official Review · Reviewer_bBjs · 2022-10-22

**Confidence:** 3
**Correctness:** 4
**Technical Novelty And Significance:** 3
**Empirical Novelty And Significance:** 3
**Recommendation:** 6

**Clarity, Quality, Novelty And Reproducibility:**

Quality: I have a few questions that I think could make the analysis more complete if addressed:
- Can the method still work well if instead of providing the best expert demonstration in the dataset, one in for example the 90th percentile - is selected?
- How important is it that the cosine distance is used as opposed to alternatives?

Clarity:
I think the paper is generally well written and clear. However, I have a few questions:
- In Figure 2, why are there so many fewer samples from the expert policy than the other? I understand that two trajectories may have different lengths, but it seems that there are many more from the behavior policy. Yet the matrix C is drawn fairly square.
- In 4.1, it says “When there are more than one episode of expert demonstration, we perform a top K aggregation strategy with K = 1”, but I think it would be more clear just to remove the part about the top K aggregation and say “we compute the optimal transport with respect to each episode independently and use the rewards from the expert trajectory that give the best episodic return.”


**Strength And Weaknesses:**

Strengths:
- The method is conceptually simple, and the quantitative results are encouraging – OTR+IQL is able to recover the performance of IQL using oracle rewards for many of the D4RL tasks.
- It’s quite surprising and impressive that a single good expert demonstration can be used for the expert trajectory in this setup.
- The empirical evaluation compares to sensible baselines on the D4RL benchmark.
- The paper is particularly clearly written and easy to understand!

Weaknesses:
- The introduction states that for behavior cloning, “generalization to new situations typically does not work well”, but I’m not sure I found the authors’ intuition in the paper for which aspect of OTR will improve generalization to new situations? It seems that labeling rewards based on a single demonstration will always make it more challenging to develop robust policies, as a single demonstration cannot provide signal about how to detect or recover from bad states. I’m curious if the authors could elaborate or if this would continue to work well in more challenging environments.
- I think it would be helpful to put the performance of the method in context by comparing the performance to behavior cloning on the top K% of dataset trajectories (like 1% or 10% that previous works like IQL report).


**Summary Of The Paper:**

The paper presents a method for performing imitation learning by first computing rewards using a method called Optimal Transport Reward labeling (OTR) that computes the Wasserstein distance between an expert demonstration and an unlabeled one. The rewards are then used in combination with existing offline RL algorithms, for example, implicit Q-learning (IQL). The process is made computationally efficient through the use of GPU-accelerated optimal transport solvers. The method is tested on locomotion tasks from the D4RL benchmark, and compared to DemoDICE, and ORIL and UDS for computing rewards. Through additional experiments, the authors show that OTR is able to effectively use up to 10 demonstrations to boost performance.

**Summary Of The Review:**

While there are related works with a similar reward relabeling idea such as [1], this paper tackles a challenging and (as far as I am aware) novel situation of learning from a very small number of demonstrations. It demonstrates promising empirical results of being able to match the performance of offline RL algorithms when given ground truth reward functions. The paper is well written and clearly presented. Therefore I vote weak accept.

[1]: Imitation Learning from Pixel Observations for Continuous Control (Cohen et. al)

---

> ### Author Response · Authors · 2022-11-14
> **Response to Reviewer bBjs**
>
> We appreciate Reviewer bBjs for their review.
>
> > I think it would be helpful to put the performance of the method in context by comparing the performance to behavior cloning on the top K% of dataset trajectories (like 1% or 10% that previous works like IQL report).
>
> Thank you. We have included the top BC performance in the table.
>
> > How important is it that the cosine distance is used as opposed to alternatives?
>
> The cosine similarity was found in [1] to have good empirical performance. We didn’t not extensively study different choices of the cosine distance but leave that to future work. You can also refer to our replies to Reviewer C2wE on some of the practical advantages of using the cosine similarity.
>
> [1] Imitation Learning from Pixel Observations for Continuous Control (Cohen et. al)
>
> > In Figure 2, why are there so many fewer samples from the expert policy than the other? I understand that two trajectories may have different lengths, but it seems that there are many more from the behavior policy. Yet the matrix C is drawn fairly square.
>
> Thank you for pointing this out. The expert has fewer samples as we would like to illustrate the practical scenario where one would have fewer expert samples compared to unlabeled ones. The matrix is downsampled for aesthetics. We will clarify this in our revision.
>
> > In 4.1, it says “When there are more than one episode of expert demonstration, we perform a top K aggregation strategy with K = 1”, but I think it would be more clear just to remove the part about the top K aggregation and say “we compute the optimal transport with respect to each episode independently and use the rewards from the expert trajectory that give the best episodic return.”
>
> Thank you! We have rephrased this according to your suggestions.

---

> ### Author Response · Authors · 2022-11-14
> **Can OTR work with less optimal demonstrations.**
>
> > Can the method still work well if instead of providing the best expert demonstration in the dataset, one in for example the 90th percentile - is selected?
>
> We expect OTR to perform somewhere between the average performance in the dataset combined and the expert demonstration. Therefore, if the 90th is significantly worse than the best episode, we expect OTR to perform worse since the better episodes are different from the 90th percentile and will have lower rewards.

---

> ### Author Response · Authors · 2022-11-14
> **Additional Reply to Reviewer bBjs**
>
> Dear Reviewer bBjs,
>
> Thank you once again for your encouraging reviews that value the simplicity and clarity of our approach, we would like to know if there are any additional things you would like to see to consider increasing your score.

---

### Official Review · Reviewer_C2wE · 2022-10-22

**Confidence:** 4
**Correctness:** 3
**Technical Novelty And Significance:** 2
**Empirical Novelty And Significance:** 2
**Recommendation:** 5

**Clarity, Quality, Novelty And Reproducibility:**

The paper writing is pretty satisfactory, and the authors explained their idea clearly. However, I have concerns regarding the novelty of this work.

The authors had better justify the benefit of this method more comprehensively. The proposed method directly applied the Wasserstein distance (Sinkhorn algorithm) to the empirical distribution of the spaces. However, it seems that the proposed reward annotating method ignores the transition kernel, which is an important ingredient for RL. MDP is a decision process, how does this method take the information from transition kernels into account?
In the contrast, PWIL proposed to find the coupling according to the sequential orders. In addition, in UDS, the authors claimed that, under certain conditions, setting rewards to zero still improves an offline RL task and provided theoretical justifications. Is it possible to elaborate more about how the proposed method can leverage the geometric information with OT?


**Details Of Ethics Concerns:**

No ethic concerns.

**Strength And Weaknesses:**

Pros:
- Overall, the paper is well-written and quite easy to follow.
- The idea of combining optimal transport with reinforcement learning is pretty novel. Given that there are only limited previous studies.
- The experimental results are encouraging.

Cons:

- The novelty is limited, considering some previous studies. I have included more detailed questions in the next section.

- In Figure 2, the cost matrix and the optimal coupling do not seem to correspond. In addition, the authors may want to annotate $s^e_3$ and $s^e_4 $in the leftmost figure.

- The authors used cosine similarity as the ground metric. Is there a principle to choose the ground cost since this is very important for the OT objective? The authors may want to refer to bisimulation metrics.

- One of the pitfalls of offline RL approach is the distribution shift problem. However, the authors did not discuss how the proposed method can handle this issue. When the expert trajectories state distribution is far away from the unlabeled ones, I am wondering whether the proposed method will still annotate the closest states by high rewards.



**Summary Of The Paper:**

In this paper, the authors focused on the offline reinforcement learning setting where a small set of expert demonstrations and a larger dataset without reward are available.
The proposed method labels the reward for the larger dataset with optimal transport theories. Specifically, the authors considered the distribution over the state space and use the optimal coupling to decide the closeness of states.
The unlabeled trajectory will be assigned a higher reward if its state distribution is closer to the state distribution from experts' data. The authors demonstrated the effectiveness of this method in several environments


**Summary Of The Review:**

The paper is overall well-written. However, I would suggest the authors elaborate more on the benefits of labeling rewards with optimal transport coupling.

---

> ### Author Response · Authors · 2022-11-14
> **Response to Reviewer C2wE**
>
> We appreciate Reviewer C2wE for their review of our manuscript.
>
> > The authors used cosine similarity as the ground metric. Is there a principle to choose the ground cost since this is very important for the OT objective? The authors may want to refer to bisimulation metrics.
>
> A few previous works (e.g., [1,2]) applying OT for imitation learning have used different cost functions. Defining a metric that can be used to measure the similarity of trajectories is not trivial as noted by [1,4]. Nevertheless, the cosine similarity forms an important ingredient for measuring the similarity between MDPs in previous work [3,4].
>
> From a practical perspective, the choice of the cost function can influence the reward scale, which can have a great impact on some offline RL backbones. In this case, the cosine similarity is advantageous since, unlike the Euclidean metric, the cosine similarity is bounded and is not sensitive to the magnitude of the inputs.
>
> [1] Primal Wasserstein Imitation Learning (Dadashi et. al)
>
> [2] Imitation Learning from Pixel Observations for Continuous Control (Cohen et. al)
>
> [3] A taxonomy for similarity metrics between Markov decision processes (Garcia et. al)
>
> [4] Metrics for Finite Markov Decision Processes (Ferns et. al)

---

> > ### Comment · Reviewer_C2wE · 2022-12-02
> > **Reply**
> >
> > I would like to thank the authors for their reply. I must say the idea of incorporating OT with offline RL is quite novel and I do appreciate the concept. However, my questions are not fully resolved, hence I would like to summarize my questions further so as to help clarify the contribution.
> >
> > (1) Figure 2 is not clear enough. The dimension of the cost matrix is wrong according to the number of samples. In addition, the cost matrix and the optimal coupling do not correspond. It is apparent that the $\mu^*$ matrix is not the optimal coupling given the cost matrix $C$. The authors may want to address these problems since figure 2 serves an important role in illustrating the ideas.
> >
> > (2) The ground cost function is very important. However, I am wondering whether the authors are claiming that cosine similarity is always the best choice over the Euclidean distance. I believe at least there should be some strategies to choose the ground metric (even by heuristics). Take the maze environment as an example [1, 2]. The two states in a maze can neither be measured by Euclidean space nor cosine similarity. Is it possible to have some experimental results in these environments?
> >
> > (3) The authors claimed, "OTR finds coupling according to sequential orders so it respects the transition kernel",  However, in the paper, the coupling is solved via the standard Sinkhorn algorithm. I think the authors may want to emphasize why just using the standard OT formulation for the states can have advantages given the presence of PWIL which designs a specialized algorithm for the coupling.
> >
> > [1] https://github.com/enlite-ai/maze
> > [2] https://github.com/zuoxingdong/mazelab

---

> > > ### Author Response · Authors · 2022-12-03
> > > **Reply to Reviewer C2wE**
> > >
> > > Dear reviewer C2wE,
> > >
> > > Thank you for your reply.
> > >
> > > > (1) Figure 2 is not clear enough. The dimension of the cost matrix is wrong according to the number of samples. In addition, the cost matrix and the optimal coupling do not correspond. It is apparent that the  matrix is not the optimal coupling given the cost matrix . The authors may want to address these problems since figure 2 serves an important role in illustrating the ideas.
> > >
> > > Thank you for raising this. We have now updated the plot, so the policy and demonstration have the same number of samples. Unfortunately we are unable to to update the manuscript at this point as the rebuttal period has passed. We are sorry that the previous plot has caused some confusion.
> > >
> > > In addition, we have shared a Colab for reproducing the new figure. Hopefully, this can serve to clarify the process further and preview the updated figure.
> > > https://colab.research.google.com/drive/1YCrSNbLZ_QylZzHcddoh-Gld-ye2YjwO?usp=sharing
> > >
> > > > The ground cost function is very important. However, I am wondering whether the authors are claiming that cosine similarity is always the best choice over the Euclidean distance. I believe at least there should be some strategies to choose the ground metric (even by heuristics). Take the maze environment as an example [1, 2]. The two states in a maze can neither be measured by Euclidean space nor cosine similarity. Is it possible to have some experimental results in these environments?
> > >
> > > We agree with the reviewer that the cost function is very important. We _do not_ claim that cosine similarity is always the best choice. It is a design choice that we found to give consistent performance when the state space consists of proprioceptive states. However, they would not be good ground costs if the observations are pixels which motivate [1, 2] to consider additional representation learning either by CTC or a target encoder. Due to limited time we are unfortunately not able to evaluate on the two maze environments that you mentioned. However, since OTR is simple to implement, we hope we or someone else can try it out in these environments at some point.
> > >
> > > > (3) The authors claimed, "OTR finds coupling according to sequential orders so it respects the transition kernel", However, in the paper, the coupling is solved via the standard Sinkhorn algorithm. I think the authors may want to emphasize why just using the standard OT formulation for the states can have advantages given the presence of PWIL which designs a specialized algorithm for the coupling.
> > >
> > > We respectfully disagree with the reviewer that PWIL [2] respects the transition kernel whereas OTR doesn't. To the best of our knowledge, our formulation is the same as PWIL. In fact, our formulation (Eqn 2) is almost the same as in PWIL's Eqn 4 except that we are solving the OT problem with respect to the behavior policy in the dataset (which are approximated with the empirical distributions as the episodes in the dataset). To our understanding, the motivation behind the specialized algorithm (Alg 1) proposed in PWIL is just a greedy approximation that upper bounds the solution of Eqn 4. Alg 1 is convenient in the online setting as it allows computing a scalar reward at every time step without having to wait for an episode to finish. However, this is unnecessary in the offline setting since we have the episodes a priori. In addition, the solution from PWIL specialized algorithm can be worse than solving the true OT problem (e.g., via the Sinkhorn algorithm but with very small entropic regularization). For us, we simply solve Eqn 4 via the Sinkhorn algorithm which is a principled approach to obtaining the optimal coupling.
> > >
> > > We hope our follow-up answers your questions. Please let us know if anything is still unclear.
> > >
> > > ----
> > > [1] Imitation Learning from Pixel Observations for Continuous Control (Cohen et. al)
> > >
> > > [2] Primal Wasserstein Imitation Learning (Dadashi et. al)

---

> ### Author Response · Authors · 2022-11-14
> **Issues of Distribution Shifts**
>
> > One of the pitfalls of offline RL approach is the distribution shift problem. However, the authors did not discuss how the proposed method can handle this issue. When the expert trajectories state distribution is far away from the unlabeled ones, I am wondering whether the proposed method will still annotate the closest states by high rewards.
>
> We agree with the reviewer that mitigating distribution shift is one of the most important problems in offline RL research. In this paper this is handled by our choice of offline RL backbone - IQL - which incorporates implicit policy constraints and avoids querying OOD actions for critic learning to minimize the effect of distribution shifts.
>
> The final cost (negative of reward) depends on both the cost function between states and the optimal transport plan. The optimal transport plan is typically sparse and a state in the unlabeled episode will transport the majority of the masses to states in the expert demonstration that are closest with respect to the cost function. In this case, the final reward is largely dependent on the distance between the unlabeled state with the closest state in the expert demonstration. This is illustrated in Figure 2 where states far from the expert trajectory are labeled with lower rewards.

---

> ### Author Response · Authors · 2022-11-14
> **MDP Transition Kernel**
>
> > The authors had better justify the benefit of this method more comprehensively. The proposed method directly applied the Wasserstein distance (Sinkhorn algorithm) to the empirical distribution of the spaces. However, it seems that the proposed reward annotating method ignores the transition kernel, which is an important ingredient for RL. MDP is a decision process, how does this method take the information from transition kernels into account? In the contrast, PWIL proposed to find the coupling according to the sequential orders.
>
> Same as PWIL, OTR finds coupling according to sequential orders so it respects the transition kernel of the MDP. The difference between OTR and PWIL is that OTR uses the Sinkhorn solver instead of the greedy approximation used by PWIL.

---

> ### Author Response · Authors · 2022-11-14
> **How OTR Leverages Geometry Information**
>
> > In addition, in UDS, the authors claimed that, under certain conditions, setting rewards to zero still improves an offline RL task and provided theoretical justifications. Is it possible to elaborate more about how the proposed method can leverage the geometric information with OT?
>
> Regarding UDS, the UDS authors suggest that UDS is effective when the unlabeled trajectories are all suboptimal ) because the 0 rewards are then, in some sense, correct labels. However, UDS is not effective when the unlabeled dataset also contains high-quality demonstrations (as discussed by the UDS paper authors). We confirm this in our evaluation (see e.g., the results of IQL+UDS on medium-v2 or medium-expert datasets). This is because UDS assigns minimal rewards uniformly to all unlabeled data even when they are potentially high-quality (see e.g. Figure 4 (a)).
>
> In comparison, by leveraging the Wasserstein distance between the unlabeled trajectories with the expert demonstrations, OTR will assign high rewards to unlabeled episodes when they are close to the expert demonstrations (in terms of Wasserstein distance based on a chosen cost function that takes into account the geometry of the state(-action) distributions). Therefore, OTR can differentiate the qualities of the unlabeled episodes while UDS can’t.
>
> UDS should never outperform using oracle rewards for unlabeled datasets. However, it may outperform reward learning approaches when the reward estimator has a large positive bias (e.g., a NN reward model that assigns high rewards to transitions that are considered bad by an oracle reward function.) We believe that OTR outperforms alternatives (UDS, ORIL) because the rewards predicted by OTR have a lower bias compared to other reward learning algorithms and can be used for differentiating qualities of data in the unlabeled dataset.

---

> ### Author Response · Authors · 2022-11-14
> **Additional Reply to Reviewer C2wE**
>
> Dear Reviewer C2wE,
>
> Thank you for your review.  We hope that we have addressed all of your comments, in particular how OTR addresses distribution shifts and how OTR differs from UDS. In light of this, we hope that you can consider raising your score or alternatively let us know what additional result or clarification you would find helpful.

---

### Official Review · Reviewer_heNC · 2022-10-24

**Confidence:** 4
**Correctness:** 3
**Technical Novelty And Significance:** 3
**Empirical Novelty And Significance:** 3
**Recommendation:** 6

**Clarity, Quality, Novelty And Reproducibility:**

- It is unclear to me how the cost function is defined between states exactly. The authors mentioned in the 4.1 Setup that they use the cosine distance and reference the paper by Cohen et al. (2021). However, the referenced paper deals with pixel observations and the cosine distance is computed in the encoded observation. As far as what I understand, all the experiments in the paper considers state observations which adds uncertainty on how the distance is computed. This is a very important detail as the distance function (or the cost function) directly affects how the Wasserstein distance between trajectories is defined, which is the main objective that the proposed uses to minimize to provide the shaped reward function.

- Are the expert demonstrations used for OTR also included in the dataset for the IQL oracle (the baseline with ground-truth rewards)? The most relevant sentence I could find is this -- "For each environment, we use the medium-v2, medium-replay-v2 and medium-expert-v2 datasets to construct the expert demonstrations and the unlabeled dataset." I found it to be vague as it could be interpreted as that the expert demonstrations are constructed by selecting the best trajectories from all three datasets combined for each environment. If it is indeed the case that the best trajectories are selected from all three datasets combined for each environment, I think these expert trajectories should also be added to the individual datasets (medium-v2, medium-replay-v2 and medium-expert-v2) for the IQL oracle to ensure a fair comparison since the paper tries to claim that the OTR+IQL could consistently match the performance of offline RL with ground-truth rewards (e.g., the last sentence of the abstract). Otherwise, the claim is not well-supported because both IQL oracle and OTR+IQL have advantages in their own way (IQL oracle has access to ground truth reward but possibly worse trajectories, OTR+IQL has access to possibly better trajectories but with much fewer reward labels).

**Strength And Weaknesses:**

*Strength*
- The proposed approach is conceptually simple and elegant with strong empirical results on a range of tasks.

*Weaknesses*
- OTR introduces additional hyper-parameters such as $\alpha, \beta$ in the squashing function (for reward labeling). The paper currently has no sensitivity analysis on how these parameters could affect the performance.
- There is uncertainty over how the cost between states is defined in the optimal transport formulation (see the section below), which could impact the reproducibility of the paper.
- I found the claim that the proposed approach (OTR) can consistently match the performance of offline RL with ground-truth rewards is not well-supported by the current set of experiments (see the section below).

**Summary Of The Paper:**

The paper proposes a new algorithm for reward-free offline reinforcement learning when a few expert demonstrations are available. The core idea is to assign rewards to the unlabelled trajectories based on their distance to the expert demonstrations such that the closer the trajectories they are to the expert demonstrations, the higher rewards they get assigned. The proposed approach (OTR) uses an optimal transport formulation where each trajectory is treated as a discrete distribution with uniform density over the states in the trajectory and the distance between two trajectories is defined as the Wasserstein distance between these two discrete distributions. The paper demonstrates that OTR can be combined with existing offline RL methods with little implementation and computation overhead to tackle reward-free offline RL with demonstrations and often matches the performance of offline RL algorithms when rewards are given on a range of MuJoCo locomotion tasks and Adroit manipulation tasks.

**Summary Of The Review:**

The paper is generally well-written with strong empirical results. Although I have some concerns over the reproducibility of the work and empirical comparisons with respect to the baselines (see above), they do not occur to me to be major issues as long as they are addressed appropriately. Overall, I would recommend acceptance of the paper (borderline).

---

> ### Author Response · Authors · 2022-11-14
> **Response to Reviewer heNC**
>
> We appreciate reviewer heNC for their kind comments and review of our manuscript.
>
> > OTR introduces additional hyper-parameters such as $\alpha$, $\beta$ in the squashing function (for reward labeling). The paper currently has no sensitivity analysis on how these parameters could affect the performance.
>
> We use the same hyperparameters for the squashing function as those used in PWIL which was found to perform well in an online imitation learning context.
>
> We have now included additional results in the Appendix ablating the hyperparameters in the squashing function. We found that sometimes OTR did perform worse due to different alpha and beta and this is likely an issue with the reward scale since the IQL algorithm we use can be sensitive to the reward scale. Nevertheless, we do not find OTR’s performance to deteriorate with different choices of these hyperparameters.

---

> ### Author Response · Authors · 2022-11-14
> **Additional Response to Reviewer heNC**
>
> > It is unclear to me how the cost function is defined between states exactly. The authors mentioned in the 4.1 Setup that they use the cosine distance and reference the paper by Cohen et al. (2021). However, the referenced paper deals with pixel observations and the cosine distance is computed in the encoded observation. As far as what I understand, all the experiments in the paper considers state observations which adds uncertainty on how the distance is computed. This is a very important detail as the distance function (or the cost function) directly affects how the Wasserstein distance between trajectories is defined, which is the main objective that the proposed uses to minimize to provide the shaped reward function.
>
> The cosine distance doesn’t necessarily need pixels as arguments, but works with other features as well. For example, Cohen et al. (2021), use the cosine distance between embeddings computed from the encoder network. It is computed as 1 - Cosine Similarity, where
> $$
> \mbox{Cosine Similarity} = \frac{\sum_{i=1}^{n}{x_{i} y_{i}}}
>            {\sqrt{\sum_{i=1}^{n}{x_{i}^{2}}}
>             \sqrt{\sum_{i=1}^{n}{y_{i}^{2}}}}
> $$
> Since the datasets in D4RL consist of state observations, we use the cosine distance between the state observations. We hope this clarifies this point.

---

> ### Author Response · Authors · 2022-11-14
> **Fairness of Comparison with IQL using Oracle Rewards**
>
> > Are the expert demonstrations used for OTR also included in the dataset for the IQL oracle (the baseline with ground-truth rewards)?
>
> Yes, otherwise the comparison with IQL oracle would be unfair.
>
> To be more specific, for example, consider halfcheetah-medium-replay-v2, for OTR, we will select the expert demonstrations as the episodes with the best episode return (computed from the dataset rewards). Then for each episode in halfcheetah-medium-replay-v2, we will run OTR to compute the labeled rewards with respect to the best episodes that we have just selected from the same dataset.
> In this case, the dataset used by OTR+IQL is identical to the dataset used by the IQL oracle (except for the rewards), allowing for easy evaluation of the effect of using labeled instead of oracle rewards.

---

> ### Author Response · Authors · 2022-11-14
> **Additional Reply to Reviewer heNC**
>
> Thank you for your review. We hope that we have clarified the confusion we made in our submission about the evaluation procedure and details on the distance function. We hope our clarification would allow you to consider increasing your score in light of this or alternatively help us understand what is missing that would prevent you from doing so.

---

### Official Review · Reviewer_eZXx · 2022-11-04

**Confidence:** 2
**Correctness:** 3
**Technical Novelty And Significance:** 1
**Empirical Novelty And Significance:** 1
**Recommendation:** 5

**Clarity, Quality, Novelty And Reproducibility:**

### Clarity
- The submission is clearly written, and their contribution is well-summarized.
- All mathematical notations seem correct.

### Quality
- Various experiments were done.
- Experiment results are well-summarized.

### Novelty
- The submission can be regarded as a simple offline extention of PWIL.
- Most of ideas regarding OT (e.g., using exponentiated rewards, computing optimal alignment with solver) are already presented by PWIL.

### Reproducibility
- Source code is not given, but hyperparameters are shared, which seems to be reproducible.

**Strength And Weaknesses:**

### Strength

- The proposed algorithm outperforms baselines.

### Weaknesses

- The contribution seems incremental. Using OT to acquire IL rewards was proposed already by PWIL (by Dadashi et al) although the idea was for online imitation learning. The submission extends PWIL's idea to offline IL by using offline RL instead.
- The complexity of using OT is not well-described. When the number of expert demonstrations is large, the reward estimation may not be scalable due to solver's complexity.


**Summary Of The Paper:**

Authors propose an offline imitation learning algorithm that annotates rewards by using Optimal Transport (w.r.t. expert trajectories) solver and uses those reward with the offline RL to mimic expert behavior without environment interactions. The algorithm is evaluated on D4RL benchmarks (which is a widely adopted benchmark for offline RL and IL algorithms) and is shown to outperform existing offline IL algorithms.

**Summary Of The Review:**

Although the algorithm empirically outperforms baselines, the idea doesn't seem novel, and the contribution seems incremental.

---

> ### Author Response · Authors · 2022-11-14
> **Response to Reviewer eZXx**
>
> We appreciate reviewer ezXx for their review of our manuscript.
>
> ----
> **Differences from PWIL**
> > The contribution seems incremental. Using OT to acquire IL rewards was proposed already by PWIL (by Dadashi et al) although the idea was for online imitation learning. The submission extends PWIL's idea to offline IL by using offline RL instead.
>
> We agree with you that PWIL is a very important paper that proposes many relevant ideas for the optimal transport formulation of imitation learning. The idea of leveraging the Wasserstein distance as a reward function is used extensively for follow-up work that also uses Optimal Transport for imitation learning.
>
> Our work differs from PWIL in the following ways:
> 1. PWIL is a state-of-the-art online imitation learning algorithm, there is no previous study applying PWIL in offline imitation learning, which presents a unique set of challenges (e.g. the problem of distribution shifts). To the best of our knowledge, we are the **first** to consider using the optimal transport formulation of imitation learning in the offline setting. In addition, we also demonstrate that OTR is the **only** algorithm that consistently and robustly achieves good performance compared to offline RL using oracle rewards.
> 2. PWIL solves the optimal transport problem using an upper bound of the Wasserstein distance and a greedy algorithm. We use the Sinkhorn distance which is the Wasserstein distance with entropic regularization. Using the Sinkhorn distance allows us to use efficient solvers for the GPU which we demonstrate to be computationally very efficient in this regime.
> 3. We also differ from PWIL in terms of the choice of the cost function (Cosine distance vs Euclidean), and how to combine multiple demonstrations (averaging over all experts vs average over the best experts).
> 4. For all experiments used in the paper, we only use the observations from the expert demonstrations (no actions). This is useful since in practice we may not observe the actions. While PWIL can in principle work with only observations, as found in the original paper, **PWIL’s performance deteriorates without incorporating the actions**. We found that OTR works well even in the absence of actions in the expert demonstration.
>
> To understand if simply switching the RL algorithm used in PWIL to an offline RL algorithm is sufficient for good performance, we ran some experiments combining PWIL with IQL. We use the implementation from https://github.com/deepmind/acme/tree/master/acme/agents/jax/pwil.
> Please refer to our updated manuscript for the comparison between OTR and PWIL
>
> We found that we are unable to get any meaningful result with PWIL+IQL when only the expert observations are used for the reward calculation, which is the setting we used for evaluating OTR.

---

> > ### Author Response · Authors · 2022-11-14
> > **Computational Complexity of OT**
> >
> > > The complexity of using OT is not well-described. When the number of expert demonstrations is large, the reward estimation may not be scalable due to solver's complexity.
> >
> > We agree that OT does not scale particularly well with a large number of expert demonstrations. Nevertheless, we believe that having a good algorithm that works well with a small number of expert demonstrations is still very valuable for practical applications since some applications only have very few demonstrations. Also in practice, since we only have to compute the rewards _once_ over the entire dataset, we found the computational overhead to be manageable for the problems we considered in this paper.
> >
> > For cases where the number of expert demonstrations is large, we believe algorithms such as ORIL [1] which learns parametric reward functions may scale better. However, in the low-data regime, these methods may suffer from overfitting issues and hence do not perform well as we have demonstrated in the paper.
> >
> > For complexity, the time complexity for labeling one demonstration is $O(NT^2)$ instead of $O(N^2T^2)$ where $N$ is the number of expert demonstrations and $T$ is the episode length.
> > This is because we aggregate the optimal transport results from multiple experts by taking the average of OT costs over different demonstrations instead of computing the optimal coupling between samples of all expert trajectories and the unlabeled trajectories. When running on accelerators, this aggregation strategy is particularly useful since it allows us to easily parallelize the computing of the OT for each expert demonstration independently.
> >
> > [1] Offline Learning from Demonstrations and Unlabeled Experience [Zolna el al.]

---

> ### Author Response · Authors · 2022-11-14
> **Additional Reply to Reviewer eZXx**
>
> > Source code is not given, but hyperparameters are shared, which seems to be reproducible.
>
> We are in the process of preparing the code to be fully open-sourced for reproducing the results in the paper.
>
> Dear Reviewer eZXx, thank you for your review. We hope that we have clarified the contributions we made in our paper and the relationship to previous work. We would be grateful if you would consider increasing your score in light of this or alternatively help us understand what is missing that would prevent you from doing so.

---

### Author Response · Authors · 2022-11-14
**General Rebuttal from the Paper Authors**

Dear reviewers, we would like to thank the reviewers for their thorough reviews.

To summarize, we present Optimal Transport Reward (OTR) labeling for annotating unlabeled offline RL datasets with reward signals given a small number of expert demonstrations. Our approach is inspired by previous work in online imitation learning that interprets the Wasserstein distance as a reward signal for RL.

Reviewer heNC commented that our approach is _“conceptually simple and elegant”_. Reviewer bBjs commented that our approach is very _“easy to understand”_ Despite the simplicity, we demonstrate that our simple algorithm outperforms a few previous works on offline imitation learning, allowing us to recover the performance of the offline RL algorithm using oracle reward functions.
In addition, we show that our algorithm is computationally fast compared to alternatives. The simplicity and strong empirical performance are the main strengths of our paper.

We have updated the manuscript incorporating some of the suggestions made by all reviewers. In addition, we have now included additional evaluation related to PWIL and ablation on some hyperparameters of OTR.

We will address individual comments from the reviewers by replying to separate threads below.

---

### Decision · Program_Chairs · 2023-01-20

**Decision:**

Accept: notable-top-25%

**Justification For Why Not Higher Score:**

With a deeper discussion of appropriate costs, and without the concerns that multiple reviewers had about whether the algorithm was sufficiently novel, I would have recommended accept.



**Justification For Why Not Lower Score:**

I think a talk that highlights this paper would be a valuable part of the conference; I think a spotlight would be enough to get the main ideas across.



**Metareview: Summary, Strengths And Weaknesses:**

(a) Summary: Given a dataset of trajectories, a subset of which are known to be expert-generated, but none of which include reward information, this paper proposes an algorithm for assigning  to unlabelled trajectories with the goal of assigning higher rewards to trajectories that are closer to expert demonstrations.  Existing offline RL methods can then be used to learn a policy based on the estimated rewards.

(b) Strengths: All reviewers agreed that this is a conceptually simple approach, and the empirical performance appears to be quite good.

(c) Weaknesses: The reviewers had two main concerns.  First, whether this is an incremental contribution over existing work.  Second, the choice of cost function appeared to be somewhat arbitrary; a more thorough discussion of this design decision would have made for a much stronger paper.



**Note From Pc:**

if the above contains the word "oral" or "spotlight" please see: "oral" presentation means -> notable-top-5% and "spotlight" means -> notable-top-25%. As stated in our emails, we are disassociating presentation type from AC recommendations

**Summary Of Ac-Reviewer Meeting:**

Reviewer C2wE,  Reviewer heNC, Reviewer bBjs, and I met on November 30 to discuss the paper. Reviewer eZXx had a life emergency that precluded his attendance.

 Reviewer heNC was the most positive reviewer.  In his view the paper was clearly above the bar; the method proposed by the paper is simple and seems to have beneficial results.  Reviewer bBjs was similarly positive, but was not willing to be the lead advocate for the paper.  His major concern was whether the algorithm was really doing the right thing.

There was broad agreement that the choice of costs is not adequately examined in the paper; this is a very important decision, and it is easy to construct a scenario where the paper's choice will break down (e.g., a maze).

There was also some question about whether this is a somewhat incremental contribution beyond PWIL.

In the end we agreed that if the costs were considered more carefully, this would be an easy accept decision.

In my view, a method doesn't have to be perfectly general to be a valuable contribution.  The appropriate choice of costs depends on the properties of the environment and may need to be reasoned about on a case-by-case basis.  The paper would be stronger if it provided some insight into how to do that reasoning, but after the discussion I leaned toward acceptance.